# Stereochemistry of Astaxanthin Biosynthesis in the Marine Harpacticoid Copepod *Tigriopus Californicus*

**DOI:** 10.3390/md18100506

**Published:** 2020-10-05

**Authors:** Alfonso Prado-Cabrero, Ganjar Saefurahman, John M. Nolan

**Affiliations:** 1Nutrition Research Center Ireland, School of Health Science, Carriganore House, Waterford Institute of Technology West Campus, Carriganore, Waterford X91K236, Ireland; ganjars@apps.ipb.ac.id (G.S.); jmnolan@wit.ie (J.M.N.); 2Surfactant and Bioenergy Research Center, IPB University, Bogor 16143, Indonesia

**Keywords:** astaxanthin, stereochemistry, *Tigriopus*, copepod, *meso*-zeaxanthin, zeaxanthin, lutein

## Abstract

The harpacticoid copepod *Tigriopus californicus* has been recognized as a model organism for the study of marine pollutants. Furthermore, the nutritional profile of this copepod is of interest to the aquafeed industry. Part of this interest lies in the fact that *Tigriopus* produces astaxanthin, an essential carotenoid in salmonid aquaculture. Here, we study for the first time the stereochemistry of the astaxanthin produced by this copepod. We cultured *T. californicus* with different feeding sources and used chiral high-performance liquid chromatography with diode array detection (HPLC-DAD) to determine that *T. californicus* synthesizes pure 3*S*,3’*S*-astaxanthin. Using *meso*-zeaxanthin as feed, we found that the putative ketolase enzyme from *T. californicus* can work with β-rings with either 3*R*- or 3*S*-oriented hydroxyl groups. Despite this ability, experiments in the presence of hydroxylated and non-hydroxylated carotenoids suggest that *T. californicus* prefers to use the latter to produce 3*S*,3’*S*-astaxanthin. We suggest that the biochemical tools described in this work can be used to study the mechanistic aspects of the recently identified avian ketolase.

## 1. Introduction

*Tigriopus* is a genus of harpacticoid copepods that inhabits rocky tide pools in most of the planet’s coasts [1]. Due to the wide geographical distribution and ease of cultivation of this crustacean, *Tigriopus* is becoming a model organism for evaluating the impact of pollutants on the marine ecosystem [2]. The high nutritional value of *Tigriopus* has also attracted the interest of aquaculture, and this copepod has been tested for aquafeeds [3,4,5].

Part of the nutritional interest of this copepod lies in its reddish coloration, since *Tigriopus* spp. produces astaxanthin [6]. This carotenoid, sourced from chemical synthesis mainly, is used to support the healthy growth of farmed salmon and trout [7,8], and is also essential for these species to acquire the characteristic reddish hue that is crucial for customer acceptance [9]. On the other hand, astaxanthin from microalgae origin is marketed for direct human consumption in capsules [10], as experiments in murine models suggest that this carotenoid may be beneficial in cognition [11], cardiac function [12], and inflammation [13]. Nevertheless, a recent meta-analysis of the randomized placebo-controlled trials conducted with astaxanthin suggests that more work is needed to confirm these benefits in humans [14].

Genetic studies in birds have contributed the most prominent advances in the study of astaxanthin biosynthesis in animals with the identification of the ketolase gene [15,16]. However, biochemical studies in zooplankton have also contributed to the knowledge of this pathway [17,18,19,20]. Along this line, Weaver et al. recently proposed *T. californicus* as a model organism for the study of astaxanthin synthesis in animals [18].

Both crustaceans and birds that produce astaxanthin use ingested carotenoids as precursors. From these precursors, β-carotene, echinenone, and canthaxanthin are non-hydroxylated carotenoids, while β-cryptoxanthin, zeaxanthin, and lutein have hydroxyl groups. β-carotene, a major carotenoid in microalgae, is believed to be the usual precursor of astaxanthin in crustaceans, whereas zeaxanthin is pointed out as the preferred precursor for birds [15]. Hydroxylated carotenoids potentially determine the stereochemistry of astaxanthin produced, while the use of non-hydroxylated carotenoids leaves the stereochemistry of astaxanthin to the mechanism of action of the β-hydroxylase. This, together with the possibility of ingestion and accumulation of astaxanthin of different stereochemistry than that produced, can finally yield one or more of the following stereoisomers: 3*S*,3’*S*-astaxanthin, 3*R*,3’*S*/3’*S*,3*R*-(*meso*)-astaxanthin or 3*R*,3’*R*-astaxanthin (Figure 1). Of note, the stereochemistry of astaxanthin has not yet been studied in birds. In crustaceans, the species studied yielded different stereoisomeric compositions [21,22,23].

The importance of knowing the stereochemistry of astaxanthin in microorganisms and animals relies on its potential use as a tool to identify the origin of a feeding source, as proposed for wild and farmed salmon [24,25]. This knowledge is also essential to introduce a novel food containing astaxanthin in the market, and can be important in terms of health benefits, since it seems that certain stereoisomers of astaxanthin are superior in this sense, as suggested by seminal studies in *C. elegans* [26].

In this work, we analyzed the stereoisomeric composition of astaxanthin in *Tigriopus californicus*, and concluded that this copepod produces pure 3*S*,3’*S*-astaxanthin. Furthermore, our results suggest the existence in *Tigriopus* of a degree of laxity in the ketolase activity, together with a strict selection of non-hydroxylated carotenoids as precursors of astaxanthin.

## 2. Results

### 2.1. Carotenoid Profile of Co-Cultured Microalgae and Tigriopus Californicus

We aimed to test the stereochemistry of astaxanthin produced by *T. californicus* in culture conditions as close as possible to nature. To do this, we co-cultured *T. californicus* with *R. lens*, *T. chui*, and *N. oceanica*, which provided a mix of potential precursor carotenoids to this crustacean. Figure 2A shows that the cultured microalgae accumulated violaxanthin as the major carotenoid (peak 1), and minor amounts of zeaxanthin (peak 4), canthaxanthin (peak 5), echinenone (peak 8), and β-carotene (peak 9). All these carotenoids are potential precursors of astaxanthin synthesis in *Tigriopus*. The carotenoid profile of *T. californicus* (Figure 2B) showed a main peak of all-*trans* astaxanthin (peak 13). The absence of this carotenoid in the microalgae indicates that *T. californicus* produced astaxanthin from ingested carotenoids, confirming previous results [18]. Appreciable amounts of violaxanthin were also observed (Figure 2B, peak 1), probably due to the presence of ingested microalgae in the gut of the zooplankton. Other minor peaks (peaks 17 and 18) comprised esterified astaxanthin. Canthaxanthin was not detected, suggesting that this carotenoid is not an intermediate in astaxanthin biosynthesis in *Tigriopus*, at least under the growth conditions tested. A significant carotenoid was peak 15, which exhibited a spectrum identical to that of echinenone from the microalgae (Figure 2A, peak 8) and echinenone authentic standard (Figure 2C). Nevertheless, the retention time of this carotenoid was shorter (13.52 min vs. 15.33 min of echinenone). These features suggest that this carotenoid has the chromophore of echinenone, but contains an additional oxygenated function that makes this carotenoid advance more rapidly in chromatography. Comparing this data with the literature, we tentatively identified this carotenoid as 3-hydroxyechinenone [27,28]. This carotenoid has been detected in other copepods that produce astaxanthin [20]. However, more work is needed to identify this carotenoid unequivocally.

### 2.2. Stereochemistry of Zeaxanthin Produced by the Microalgae Culture

As stated above, the microalgae culture used in the previous section provided four carotenoids with the potential to serve as a precursor of astaxanthin synthesis in *Tigriopus*. Among them, only zeaxanthin has the hydroxyl groups that could force the stereochemistry of astaxanthin (Figure 1). Although plants, microalgae, cyanobacteria, and non-photosynthetic bacteria produce 3*R*,3’*R*-zeaxanthin [29,30,31], we aimed to confirm this in our study. Figure 3 shows the retention time of the zeaxanthin present in the microalgae culture (Figure 3B, peak 2), which coincides with the retention time and the spectrum of 3*R*,3’*R*-zeaxanthin in the standard enantiomeric mixture (Figure 3A,C). Therefore, we conclude that *T. californicus* ingested pure 3*R*,3’*R*-zeaxanthin.

### 2.3. Stereochemistry of Astaxanthin Produced by Tigriopus Californicus

The next step consisted of determining the stereochemistry of the astaxanthin produced by *T. californicus* feeding on the microalgae described in the previous section. We decided to avoid saponification and use only free astaxanthin (roughly 70% of astaxanthin produced) as a proxy to the stereochemistry of this carotenoid in *Tigriopus*. In this way, we avoided the production of *cis* isomers that would have made the interpretation of the chiral analyses difficult. Figure 4 shows the chiral analysis of the free astaxanthin produced by *T. californicus* compared to the true astaxanthin racemic mixture. Free astaxanthin produced by *T. californicus* ran as a single peak (Figure 4B, Peak 1), with retention time and spectrum consistent with reference 3*S*,3’*S*-astaxanthin (Figure 4A,C). Peak 2 in Figure 4B showed a retention time compatible with 3*R*,3’*R*-astaxanthin (Figure 4B), but the spectrum of this carotenoid was clearly different from that of the reference (Figure 4C), and probably corresponds to a *cis*-isomer of astaxanthin. This result suggests that in this experiment, despite having an hydroxylated carotenoid available such as 3*R*,3’*R*-zeaxanthin, *Tigriopus* preferred to use a non-hydroxylated precursor such as β-carotene, echinenone, and/or canthaxanthin to synthesize astaxanthin. An alternative hypothesis, although less likely, is that *Tigriopus* dehydroxylated 3*R*,3’*R*-zeaxanthin, re-hydroxylated it as 3*S*,3’*S*-zeaxanthin, and finally used it to produce 3*S*,3’*S*-astaxanthin.

### 2.4. Astaxanthin Biosynthesis in Tigriopus Californicus Using Meso-Zeaxanthin as Precursor

Interestingly, Weaver et al. reported that *T. californicus* could use zeaxanthin to produce astaxanthin [18]. However, the absolute configuration of the synthesized astaxanthin was not investigated in that study. We wondered whether *Tigriopus* just adds the ketone groups to 3*R*,3’*R*-zeaxanthin or replaces the hydroxyl groups to produce 3*S*,3’*S*-astaxanthin. To answer this question, we cultured *Tigriopus* with marigold powder rich in *meso*-zeaxanthin and traces of 3*R*,3’*R*-zeaxanthin and lutein, but free of β-carotene and other non-hydroxylated precursor carotenoids, as shown in Figure 5.

Figure 6B shows how *T. californicus*, fed with the yeast reference diet, accumulated 3*S*,3’*S*-astaxanthin (Figure 6B, Peak 1), probably because the washing period was insufficient for this carotenoid to disappear from the zooplankton. When *Tigriopus* fed on marigold powder, congruently, this peak also appeared (Figure 6C, Peak 2). Nevertheless, a peak corresponding to *meso*-astaxanthin also appeared in the chromatogram (Figure 6C, Peak 3). With this result, we cannot rule out that *T. californicus* converted a percentage of *meso*-zeaxanthin into 3*S*,3’*S*-astaxanthin. However, we can confirm that *T. californicus* added a ketone group to both β-rings of *meso*-zeaxanthin and converted this carotenoid into *meso*-astaxanthin.

### 2.5. Testing Lutein as a Precursor for Astaxanthin Biosynthesis in Tigriopus Californicus

Next, we investigated the possibility that *Tigriopus* used lutein as a precursor to produce astaxanthin, as the use of this carotenoid could not be ruled out in previous studies and this pathway has been proposed for other crustaceans [18,32]. To do this, we took advantage of the fact that to use lutein, *Tigriopus* must first convert the ε-ring of this carotenoid into a β-ring by relocating a double bond (Figure 1). The result of this conversion is the production of *meso*-zeaxanthin, whose use by *Tigriopus* to produce astaxanthin is proven. Therefore, if lutein is a precursor of choice for *Tigriopus*, we should observe the appearance of *meso*-astaxanthin. To test this hypothesis, we cultivated *T. californicus* with an unidentified species of microalgae that appeared spontaneously in our outdoor facilities and produced high relative amounts of lutein (Figure 7A, peak 2) and lower amounts of canthaxanthin (peak 4) and β-carotene (peak 8). Again, *T. californicus* produced fully *trans* astaxanthin as the primary carotenoid (Figure 7B, peak 9), representing 49% of the total carotenoids produced. It is important to note that with this species of microalgae as feed, we were able to detect again the carotenoid that we tentatively identified as 3-hydroxyechinenone in Section 2.2 (Figure 7B, Peak 11).

### 2.6. Stereochemistry of Astaxanthin in Tigriopus Californicus Feeding on Lutein-Rich Microalgae

Chiral analysis of astaxanthin produced under the culture conditions described in the previous section shows that *T. californicus* produced only 3*S*,3’*S*-astaxanthin (Figure 8). The complete absence of meso-astaxanthin in the chromatogram suggests that T. californicus did not use dietary lutein to produce astaxanthin. The 3*S*,3’*S*-astaxanthin produced by Tigriopus probably comes from one of the non-hydroxylated carotenoids ingested (β-carotene or canthaxanthin). It is important to note that this experiment does not rule out whether *T. californicus* can convert lutein to astaxanthin if this is the only carotenoid available in the diet; importantly, this point is not possible to address, as any current commercial source of lutein contains zeaxanthin at around 5% (*w/w*). Nevertheless, as there was no *meso*-astaxanthin in the carotenoid extract, our results suggest that *T. californicus* prefers to use non-hydroxylated dietary carotenoids instead of lutein when they are available.

## 3. Discussion

In this work, we determined that *T. californicus* produced 3*S*,3’*S*-astaxanthin when fed on microalgae that provided carotenoids with and without hydroxyl groups as potential precursors. Then, we used stereochemistry as a readout to better understand the synthesis pathway of this carotenoid in *T. californicus*. For example, Weaver et al. reported that *T. californicus* can use zeaxanthin as a precursor. Strikingly, in our experiment with a variety of precursors including 3*R*,3′*R*-zeaxanthin, we did not detect the formation of 3*R*,3’*R*-astaxanthin. To rule out that *T. californicus* reorients the hydroxyl groups of zeaxanthin to 3*S*,3’*S*, we fed this copepod with 3*R*,3’*S*-(*meso*)-zeaxanthin. In this experiment, *T. californicus* produced *meso*-astaxanthin, indicating that the putative ketolase of this copepod can work with β-rings bearing either 3*S*- or 3*R*-hydroxyls. Thus, the absence of 3*R*,3’*R*-astaxanthin when 3*R*,3’*R*-zeaxanthin was available suggests that *T. californicus* does not choose this carotenoid as a precursor to producing astaxanthin when other precursors are available. We applied the same method to study lutein as a precursor, and again, the results indicated that this carotenoid is also not a preferred choice for *T. californicus*.

The ability of *T. californicus* to add ketone groups to carotenoids bearing hydroxyl groups oriented at either *R* or *S* is surprising. Still, it is even more surprising that *T. californicus* discards hydroxylated carotenoids if non-hydroxylated carotenoids such as β-carotene or canthaxanthin are available in the diet. This suggests that a mechanism of selection to refuse hydroxylated carotenoids as precursors of astaxanthin is operating in *T. californicus*.

Although *T. californicus* can potentially use both β-carotene and canthaxanthin as a precursor [18], in this work, we have not studied which of these carotenoids *T. californicus* preferentially uses. In either case, the results of Weaver et al. suggest that not only the putative ketolase is lax in carotenoid acceptance, but the putative β-hydroxylase of *T. californicus* is also flexible, as it can act on both ketolated and non-ketolated carotenoids.

We propose two reasons why *T. californicus* might prefer to introduce hydroxylations in carotenoids itself instead of taking advantage of available hydroxylated carotenoids such as zeaxanthin. The first one is that the 3*S* orientation of hydroxyls in astaxanthin is more advantageous in physiological terms for this copepod. However, neither the presence of *meso*-astaxanthin nor the total absence of carotenoids in its tissues seems to harm this copepod under normal laboratory conditions. The second possible reason has to do with energy metabolism, and perhaps with phenotypic displays of this crustacean. Recently, Hill et al. [33] proposed that the signal honesty hypothesis in birds, whereby a greater accumulation of astaxanthin in the feathers increases the male’s chances of being chosen by a female to procreate, has its molecular basis in a link between the synthesis of this carotenoid and mitochondrial metabolism. Hill et al. proposed that the addition of the ketone groups in the mitochondria is facilitated by good oxidative metabolism. Thus, the particular fitness level of a male would translate into a specific degree of redness in his feathers. There are no studies on the subcellular localization of astaxanthin synthesis in crustaceans, nor has the relationship of this carotenoid with oxidative metabolism been investigated in these animals. It would be very interesting to know whether the synthesis of astaxanthin in *Tigriopus* is directly related to the oxidative metabolism of the cell.

Following this reasoning, the apparent preference of *T. californicus* for non-hydroxylated carotenoids makes us wonder if the same happens in birds. This is apparently not the case, since it is generally assumed that these animals use hydroxylated carotenoids as precursors of ketolated carotenoids [15,16]. However, the stereoisomeric analysis of the final product in different species of birds could provide surprises if the hydroxyls are oriented to 3*S*. This would strongly suggest that birds also have a β-hydroxylase, in addition to the ketolase already described. Feeding experiments with *meso*-zeaxanthin, available in the market in sufficient quantities to carry out these types of experiments with birds, together with β-carotene, also available, would shed light on the existence of hydroxylations in the synthesis pathway of ketolated carotenoids in birds. From the signal honesty hypothesis proposed by Hill et al., it makes sense to think that the more oxidative functions the bird adds, the more honest their signal would be.

The non-chiral analysis of the carotenoids produced by *T. californicus* also yielded useful information, which tempts us to propose a biosynthesis pathway for astaxanthin in this copepod. We did not detect canthaxanthin in *Tigriopus*, which suggests that this carotenoid is not an intermediate in the astaxanthin synthesis pathway. Instead, we noticed a carotenoid that we tentatively identified as 3-hydroxyechinenone. Mojib et al. detected this carotenoid in five of the species of copepods they studied. This data suggest that *T. californicus* uses β-carotene as a precursor for astaxanthin via 3-hydroxyechinenone, as previously suggested for other copepods [20]. Nevertheless, further work is needed to fully identify the occurrence of 3-hydroxyechinenone in *T. californicus*. The penultimate step of the route, consisting of the production of adonixanthin or adonirubin, has not been investigated in this work.

In conclusion, in addition to the findings described in the astaxanthin synthesis pathway in *T. californicus*, we have developed a method that can be used to study the functional aspects of the ketolase enzyme in crustaceans and birds. In the case of crustaceans, further work remains to be done such as the identification of the genes for the β-hydroxylase and ketolase activity and the determination of the subcellular location and characterization of the coded enzymes.

## 4. Materials and Methods

### 4.1. Organisms and Chemicals

*Tigriopus californicus* was sourced from Reefphyto Ltd. (Newport, UK). *Rhodomonas lens* was obtained from the Bigelow National Center for Marine Algae and Microbiota (Maine, USA), *Tetraselmis chui* was obtained from the Culture Collection of Algae and Protozoa (CCAP, Scottish Marine Institute, Argyll, Scotland), and *Nannochloropsis oceanica* was obtained from the Norwegian Culture Collection of Algae (NORCCA, Oslo, Norway). Dried Brewers’ yeast was provided by Biomax Ltd. (Elvington, UK). *Meso*-zeaxanthin powder (10% dry biomass) was obtained from XABC Biotech Co Ltd. (Xi’an, China). The carotenoid standards lutein, zeaxanthin, and zeaxanthin racemic mixture were purchased from CaroteneNature (Lupsingen, Switzerland). Butylated hydroxyltoluene (BHT) was purchased from Sigma-Aldrich (Arklow, Ireland). HPLC grade methyl tert-butyl ether (MTBE), water, and isopropanol were supplied by Fisher Scientific (Dublin, Ireland). HPLC grade hexane and ethanol 96% were supplied by VWR (Dublin, Ireland). Artificial Sea Salt (Tropic Marin^®^ PRO-REEF) and Evolution Aqua K1 Micro media were purchased to McGuire’s Garden Centre (Waterford, Ireland).

### 4.2. Zooplankton Culture

#### 4.2.1. Tigriopus Culture with Microalgae

*Tigriopus californicus* was cultured in a 1200 L water tank in artificial sea water at 3.5% (*w/v*) and the microalgae species *Rhodomonas lens*, *Tetraselmis chui*, and *Nannochloropsis oceanica* over 60 days using aerated K1 Micro media to allow the growth of nitrifying bacteria. Temperature was kept at 22 °C. *T. californicus* culture was initially inoculated with *R. lens* and *T. chui*. After growth of *T. californicus* to a density of circa 1200 individuals per liter at day 50, *N. oceanica* was inoculated.

#### 4.2.2. Meso-Zeaxanthin Feeding Experiment

*T. californicus* was cultured in two aquariums of 10 L capacity with artificial seawater (3.5%). Brewer’s yeast was provided as feed dissolved in artificial seawater every second day. Ammonia levels were controlled using aerated K1 Micro media and aerated nitrate traps. After two months of culture, one of the cultures was fed during the three weeks with marigold extract rich in *meso*-zeaxanthin dissolved in artificial seawater, once per week. Temperature of the cultures was kept at 22 °C.

### 4.3. Carotenoid Extraction

#### 4.3.1. Tigriopus

A total of 10–20 mg (wet weight) of *Tigriopus* was harvested and analyzed. The fresh biomass was introduced in a 15 mL polypropilene tube, and 3 mL of acetone were added. The tube was sonicated for 5 min in a sonicator bath and then 2 mL of hexane and 5 mL of aqueous NaCl 0.9% (*w/w*) were added. The tube was agitated for 20 s and centrifuged for 5 min. The upper hexane phase was transferred to a new polypropilene tube and dried in a vacuum centrifuge. The residue was resuspended in the HPLC mobile phase and analyzed.

#### 4.3.2. Microalgae

A sample of 400 mL of microalgae culture were filtered with a nylon mesh with 100 μm pores and centrifuged for 5 min in 50 mL polypropilene tubes. The supernatants were discarded and the pellets were collected in a polypropilene tube. This tube was centrifuged again for 3 min and the pellet was washed with 10 mL of dH_2_O. The tube was frozen at −80 °C and thawed three times. In a new 50 mL Falcon tube, a saponification mix consisting in 0.1 *g* of KOH, 2 mL of EtOH 96%, and 0.5 mL of dH_2_O was prepared. The saponification solution was warmed at 45 °C and added to the microalgae pellet. The sample was incubated for 5 min at 45 °C at 250 rpm. Then, 10 mL of aqueous NaCl 0.9% was added to neutralize the saponification reaction and 5 mL of hexane was used to extract the carotenoids from the saponified sample. This extraction was repeated three times. The pooled hexane fractions were washed with one volume of aqueous NaCl 0.9%, dried in a vacuum centrifuge, and re-suspended in 0.4 mL of mobile phase.

### 4.4. HPLC Analysis

#### 4.4.1. System 1, Reverse Phase Carotenoid Analysis

Carotenoid samples were separated and quantified in a HPLC 1200 Series (Agilent Technologies, Santa Clara, CA, USA), equipped with a diode array detector, quaternary pump, degasser, thermostatically-controlled column compartment, thermostatically-controlled autosampler, and a C30-reversed phase column (250 × 4.6 mm i.d., 3 μm; YMC Europe, Dinslaken, Germany) with a guard column. The flow rate was 1 mL min^-1^ with a linear gradient from 100% A, consisting of methanol:methyl tert-butyl ether:water:triethylamine 30:10:1:0.05 (*v/v*) to 20% B, consisting of methanol:methyl tert-butyl ether 1:1 (*v/v*) within 10 min, then to 100% B within 1 min. This condition was maintained for another 24 min. The solvents were returned to the starting conditions within 1 min, and the column temperature was set at 25 °C.

#### 4.4.2. System 2, Chiral Analysis of Zeaxanthin

Carotenoid samples were analyzed in an Agilent Technologies (Palo Alto, CA, USA) 1260 Series HPLC system, equipped with a Diode Array Detector (DAD, G1315C), binary pump, degasser, thermostatically-controlled column compartment, thermostatically-controlled high performance autosampler (G1367E) and thermostatically-controlled analytical fraction collector. The column used was a Daicel Chiralpak IA-3 (amylose derivative bonded on silica-gel, 250 × 4.6 mm i.d., 3 μm; Chiral Technologies Europe, Cedex, France) with a guard column. Isocratic run with hexane:isopropanol (90:10, *v/v*) at a flow rate of 0.5 mL min^−1^ was used. The column temperature was set at 25 °C.

#### 4.4.3. System 3, Chiral Analysis of Astaxanthin

For this analysis, the HPLC 1200 Series described in system 1 was used with a Pirkle L-leucine chiral column (Regis Technologies, Morton Grove, IL, USA). The method used has been previously described [21].

## Figures and Tables

**Figure 1 marinedrugs-18-00506-f001:**
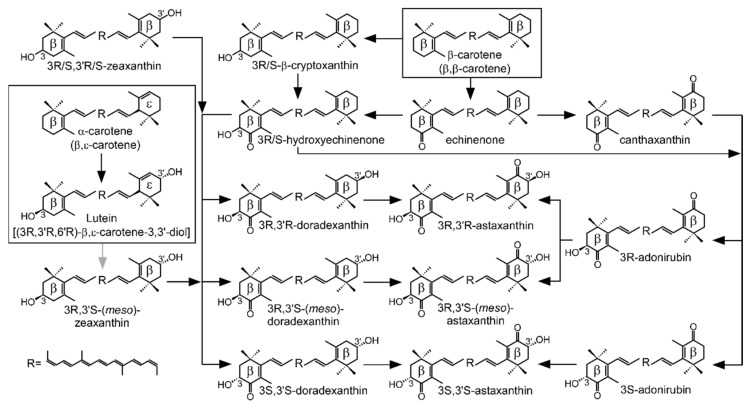
Diagram of the possible routes of astaxanthin biosynthesis in plants, microalgae, bacteria, fungi, and animals. The boxed reactions can only be carried out at the same time by plants and microalgae. Animals cannot synthesize any of the boxed carotenoids de novo. The gray arrow indicates the conversion of lutein to *meso*-zeaxanthin, an enzymatic step that has only been demonstrated at the molecular level in humans.

**Figure 2 marinedrugs-18-00506-f002:**
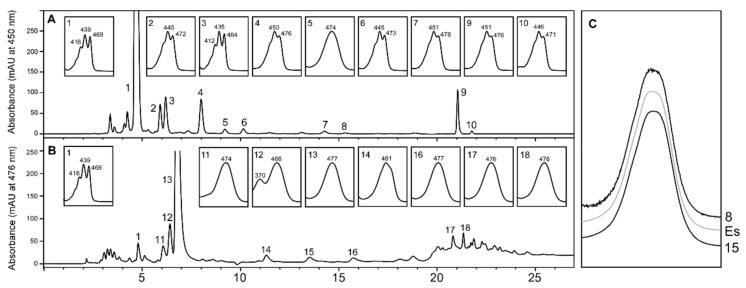
High-performance liquid chromatography (HPLC) carotenoid profile of *Nannochloropsis oceanica* and *Tigriopus californicus*. (**A**) Carotenoid profile of microalgae in reactor 6. Peak 1, violaxanthin; 4, zeaxanthin; 5, canthaxanthin; 8, echinenone; 9, all-*trans* β-carotene; 10, 9-*cis* β-carotene. (**B**) Carotenoid profile of *T. californicus* cultured in reactor 6 with *N. oceanica* as feed. Peak 1, violaxanthin; 11 and 12, *cis*-astaxanthin isomers; 13, all-*trans* astaxanthin; 14, unknown; 15, 3-hydroxyechinenone (tentative); 17 and 18, astaxanthin esters. (**C**) Spectra of echinenone standard (Es), peak 8 from panel A and peak 15 from panel B.

**Figure 3 marinedrugs-18-00506-f003:**
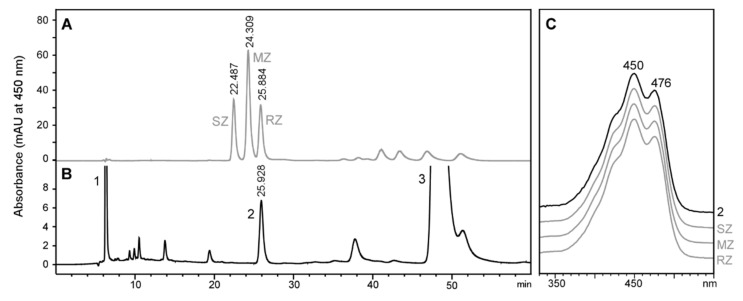
High-performance liquid chromatography (HPLC) stereoisomeric analysis of zeaxanthin present in the microalgae mixture used to culture *Tigriopus californicus*. (**A**) Authentic zeaxanthin racemic standard (peak SZ, 3*S*,3′*S*-zeaxanthin; peak MZ, *meso*-zeaxanthin, and peak RZ, 3*R*,3′*R*-zeaxanthin. (**B**) Carotenoid extract from the microalgae culture (peak 1, β-carotene; peak 2, zeaxanthin; peak 3, violaxanthin. (**C**) Spectrum of peak 2 from the HPLC chromatogram of carotenoids from *Tigriopus* and from authentic zeaxanthin racemic standard.

**Figure 4 marinedrugs-18-00506-f004:**
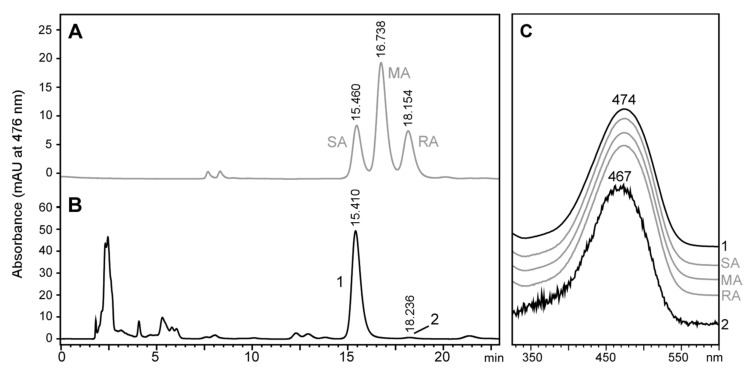
High-performance liquid chromatography stereoisomeric analysis of free astaxanthin from *Tigriopus californicus* fed with the microalgae *Nannochloropsis oceanica*. (**A**) Astaxanthin standard (racemic mixture) with the typical peaks of 3*S*,3′*S*-astaxanthin (SA), 3*R*,3′*S*-(*meso*)-astaxanthin (MA) and 3*R*,3′*R*-astaxanthin (RA). (**B**) carotenoid extract of *Tigriopus californicus*, with peak 1 showing the same retention time and spectrum of 3*S*-3′*S*-astaxanthin (SA). (**C**) Spectra of peaks from A and B.

**Figure 5 marinedrugs-18-00506-f005:**
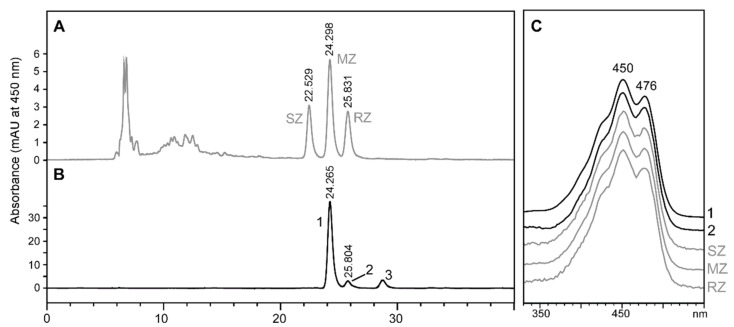
Stereoisomeric configuration of zeaxanthin in *meso*-zeaxanthin-rich marigold powder. (**A**) Zeaxanthin standard (racemic mixture) with the typical peaks of 3*S*,3′*S*-zeaxanthin (SZ), 3*R*,3′*S*-(*meso*)-zeaxanthin (MZ) and 3*R*,3′*R*-zeaxanthin (RZ). (**B**) *Meso*-zeaxanthin-rich marigold powder. Peak 1, *meso*-zeaxanhtin; peak 2, 3*R*,3′*R*-zeaxanthin and peak 3, lutein. (**C**) Spectrum of peaks identified in A and B.

**Figure 6 marinedrugs-18-00506-f006:**
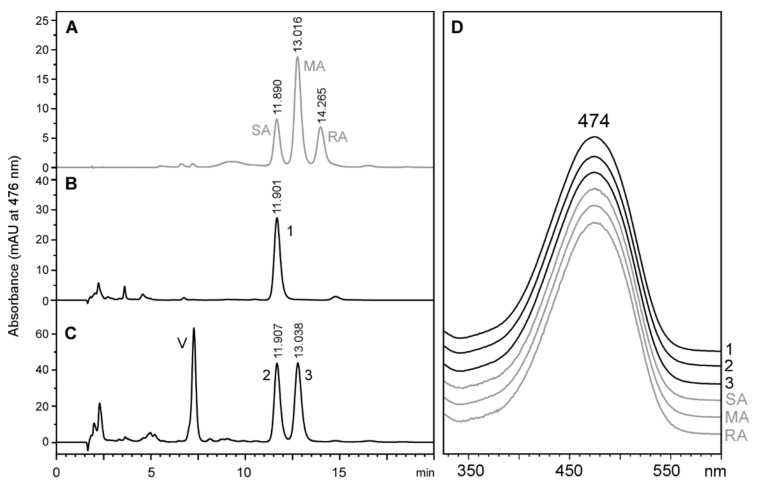
Chiral high-performance liquid chromatography (HPLC) analysis of free astaxanthin from *Tigriopus californicus* fed with *meso*-zeaxanthin. (**A**) Astaxanthin standard (racemic mixture) with the typical peaks of 3*S*,3′*S*-astaxanthin (SA), 3*R*,3′*S*-(*meso*)-astaxanthin (MA), and 3*R*,3′*R*-astaxanthin (RA). (**B**) HPLC chromatogram of the carotenoid extract of *T. californicus* fed with yeast. (**C**). HPLC chromatogram of carotenoid extract of *T. californicus* fed with marigold powder rich in *meso*-zeaxanthin. Peak V is violaxanthin. (**D**) Spectra of astaxanthin peaks detected in the HPLC analysis.

**Figure 7 marinedrugs-18-00506-f007:**
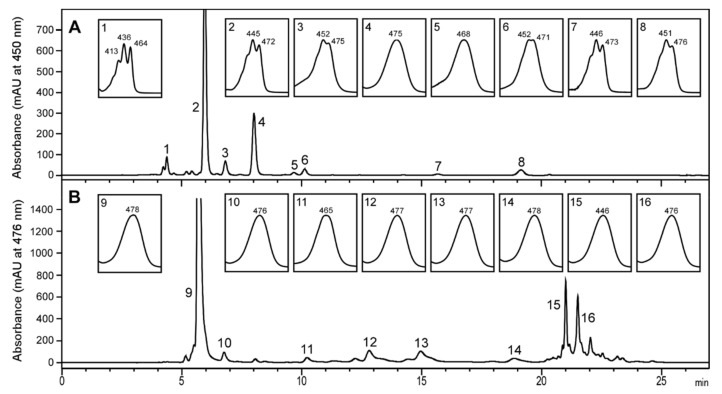
High-performance liquid chromatography (HPLC) carotenoid profile of lutein-rich microalgae and *Tigriopus californicus*. (**A**) Carotenoid profile of the microalgae. Peak 1, violaxanthin; 2, lutein; 3, unknown; 4, canthaxanthin; 5 and 6, unknown; 7, α-carotene; 8, β-carotene. (**B**) Carotenoid profile of *Tigriopus californicus* cultured with lutein-rich microalgae as feed. Peak 9, all-*trans* astaxanthin; 10, unknown; 11, 3-hydroxyechinenone (tentative); 12–14, unknown; 15 and 16, astaxanthin esters.

**Figure 8 marinedrugs-18-00506-f008:**
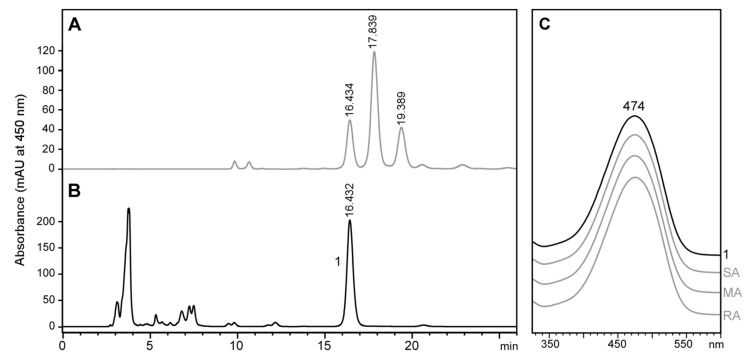
High-performance liquid chromatography (HPLC) stereoisomeric analysis of free astaxanthin from *Tigriopus californicus* fed with lutein-rich microalgae. (**A**) Astaxanthin standard (racemic mixture) with the typical peaks of 3*S*,3′*S*-astaxanthin (SA), 3*R*,3′*S* (*meso*)-astaxanthin (MA) and 3*R*,3′*R*-astaxanthin (RA). (**B**) carotenoid extract of *T. californicus*, with peak 1 showing the same retention time and spectrum of 3*S*,3′*S*-astaxanthin (SA). (**C**) Spectra of peaks from A and B.

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
