# Peer review of "Stereochemistry of Astaxanthin Biosynthesis in the Marine Harpacticoid Copepod Tigriopus Californicus"

_marinedrugs, 2020, doi:10.3390/md18100506_

Round 1

Reviewer 1 Report

In this MS, Prado-Cabrero et al. has analyzed the stereoisomeric composition of astaxanthin in Tigriopus californicus using HPLC and concluded that this copepod produces pure 3S,3'S-astaxanthin.  The authors justified the significance of this study by indicating its potential impacts in food (e.g. salmon) markets. Overall this is a nice study and the experiments are well designed and conducted. The MS is well written and is easy to follow. Analytical data are nicely analyzed and supports the conclusion. However, the structures in Fig. 1 are unreadable and NEED to be enlarged prior to its publication.

Author Response

Response to Reviewer 1 Comments

Point 1: The structures in Fig. 1 are unreadable and NEED to be enlarged prior to its publication.

Response: Dear Reviewer 1, thank you for the comments on our manuscript, and thank you very much for pointing out this issue in figure 1. To make this Figure more readable, we have increased the font size by 2 points, increased the thickness of the lines from 0.75 to 1 point, and we have further separated the double bonds from the carotenoid backbones. Please find in the attachment a comparison of the old (above) and the new Figure 1 (below).

Reviewer 2 Report

In this manuscript, the authors describe the astaxanthin synthesis pathway in T. californicus and developed a method that can be useful for the functional aspects of the ketolase enzyme in crustaceans and birds. The authors claimed that the study on the stereochemistry of the astaxanthin produced by this copepod is for the first time. This study is interesting and information could be useful for the researchers working in similar fields.  However, the manuscript is required a minor revision following the comments below.

The authors mentioned that the b-carotene is a precursor of astaxanthin in crustaceans. Since the b-carotene is a natural pigment and highly rich in the vegetables, especially carrots and spinach, the authors should explain details about the b-carotene and what exactly the functional properties in the formation of astaxanthin.

The references are not updated. Most of the references are old. The authors should collect the latest info related to their studies from existing public literature.

Author Response

Response to Reviewer 2 Comments

Point 1: The authors mentioned that the b-carotene is a precursor of astaxanthin in crustaceans. Since the b-carotene is a natural pigment and highly rich in the vegetables, especially carrots and spinach, the authors should explain details about the b-carotene and what exactly the functional properties in the formation of astaxanthin.

Response: Dear Reviewer 2, thank you for the comments on our manuscript, and thank you very much for pointing out the need of highlighting the importance of b-carotene as a precursor of astaxanthin. To point out the relevance of this carotenoid in the synthesis of astaxanthin in crustaceans, we have specified in our manuscript that b-carotene is a major carotenoid in algae, which is the main feeding source of crustaceans. Lines 47-48 have been modified as shown below:

“b-carotene, a major carotenoid in algae, is believed to be the usual precursor of astaxanthin in crustaceans”

Point 2: The references are not updated. Most of the references are old. The authors should collect the latest info related to their studies from existing public literature.

Response: Of the 33 works cited in our manuscript, four of them were published before the year 2000. However, we consider it appropriate to cite these older works due to their uniqueness, which we will detail below. We hope that the reviewer appreciates the importance of these works and the need for them to be cited in our manuscript.

Fukusho et al (1980): This work is the only one published that describes a Tigriopus culture with mass production dimensions. The production tanks described in this work contained more than 200 cubic meters of water, and one of them produced 168 kg of Tigriopus. A Tigriopus culture on such a scale has not been published since. We, therefore, believe that this work is a good example of the attempts that have been made to cultivate this copepod, as we aimed to point out in the introduction of our manuscript.

Goodwin et al (1949): We have cited this work because it was the first to describe that Tigriopus produces astaxanthin.

Foss et al (1987) and Maoka et al (1985): Strikingly, there are very few works in the literature describing the stereoisomers of astaxanthin that animals produce. These works, together with Moreti et al. (2006), are among the few existing to our knowledge, and we have cited the three of them in our manuscript.

Bartlett (1969) and Maoka et al. (1986): These works are cited along with a newer paper to support the claim that non-photosynthetic plants, microalgae, cyanobacteria, and bacteria produce 3R,3'R-zeaxanthin (Lines 108-109). Again, we want to emphasize here the scarcity of works that study the stereochemistry of carotenoids, and in this specific case, that of zeaxanthin. It is therefore that we are forced to cite the few works available on this subject, including those older.

We hope that the reviewer understands these motivations for including these older references in our manuscript. Again, we take the opportunity to thank the reviewer for highlighting this point, as this has allowed us to expose the importance of these works.

Reviewer 3 Report

In this paper, the authors studied the stereochemistry of the astaxanthin produced by T. californicus and showed that this was 3S, 3S’-astaxanthin. They also suggested the astaxanthin synthesis pathway in T. californicus. These findings are useful for the readers of Marine drugs. Therefore, I recommend the publication after minor revisions as follows. In the figure legend of the figure 2, the numbers (and carotenoid names) of some peaks were different from those in the manuscript (ex. P3, L79). In addition, there was no figure legend for the panel (C) of the figure 2. Please correct.

Author Response

Response to Reviewer 3 Comments

Point 1: In the figure legend of the figure 2, the numbers (and carotenoid names) of some peaks were different from those in the manuscript (ex. P3, L79). In addition, there was no figure legend for the panel (C) of the figure 2. Please correct.

Response: Dear Reviewer 1, thank you for the comments on our manuscript, and thank you very much for pointing out this issue in Figure 2. We have reviewed our chromatograms, and we found that the peaks of panel A were correctly marked in the text (Lines 79-80). Therefore, we have amended peak labels in the legend of Figure 2. We have also added a legend for panel C. The legend of Figure 2 now reads as shown below.

Figure 2. HPLC carotenoid profile of Nannochloropsis oceanica and Tigriopus californicus. (A) Carotenoid profile of microalgae in reactor 6. Peak 1, violaxanthin; 4, zeaxanthin; 5, canthaxanthin; 8, echinenone; 9, all-trans b-carotene; 10, 9-cis b-carotene. (B) Carotenoid profile of T. californicus cultured in reactor 6 with N. oceanica as feed. Peak 1, violaxanthin; 11 and 12, cis-astaxanthin isomers; 13, all-trans astaxanthin; 14, unknown; 15, 3-hydroxyechinenone (tentative); 17 and 18, astaxanthin esters. (C) Spectra of echinenone standard (Es), peak 8 from panel A and peak 15 from panel B.”